# A Thermo-Mechanically Robust Compliant Electrode Based on Surface Modification of Twisted and Coiled Nylon-6 Fiber for Artificial Muscle with Highly Durable Contractile Stroke

**DOI:** 10.3390/polym14173601

**Published:** 2022-08-31

**Authors:** Sungryul Yun, Seongcheol Mun, Seung Koo Park, Inwook Hwang, Meejeong Choi

**Affiliations:** 1Tangible Interface Creative Research Section, Electronics and Telecommunications Research Institute (ETRI), Daejeon 34129, Korea; 2Human Enhancement & Assistive Technology Research Section, Electronics and Telecommunications Research Institute (ETRI), Daejeon 34129, Korea

**Keywords:** surface modification, electro-thermal, contractile strain, actuation durability

## Abstract

In this paper, we propose a novel and facile methodology to chemically construct a thin and highly compliant metallic electrode onto a twisted and coiled nylon-6 fiber (TCN) with a three-dimensional structure via surface modification of the TCN eliciting gold-sulfur (Au-S) interaction for enabling durable electro-thermally-induced actuation performance of a TCN actuator (TCNA). The surface of the TCN exposed to UV/Ozone plasma was modified to (3-mercaptopropyl)trimethoxysilane (MPTMS) molecules with thiol groups through a hydrolysis-condensation reaction. Thanks to the surface modification inducing strong interaction between gold and sulfur as a formation of covalent bonds, the Au electrode on the MPTMS-TCN exhibited excellent mechanical robustness against adhesion test, simultaneously could allow overall surface of the TCN to be evenly heated without any significant physical damages during repetitive electro-thermal heating tests. Unlike the TCNAs with physically coated metallic electrode, the TCNA with the Au electrode established on the MPTMS-TCN could produce a large and repeatable contractile strain over 12% as lifting a load of 100 g even during 2000 cyclic actuations. Demonstration of the durable electrode for the TCNA can lead to technical advances in artificial muscles for human-assistive devices as well as soft robots those requires long-term stability in operation.

## 1. Introduction

Recently, as a new class of artificial muscle, twisted and coiled polymer actuators (TCPAs) have given much attention due to their excellent mechanical properties, high flexibility, cost-effectiveness, and large thermally-induced contractile deformation exceeding stroke of human skeletal muscles as simultaneously enabling periodic lifting a load over 20 MPa even under a light-weight structure [1]. Many researchers have introduced polymer artificial muscles possessing the structural resilience using various fibers, such as shape memory polymer [2], polymer bimorphs [3], polymer nanocomposites with carbon nanotube (CNT) or graphene oxide [4], natural rubbers [5], and hydrogels [6]. Technical advances in actuation mechanism, materials, and fabrication process enables the TCPAs to be utilized in prospective application fields of prosthetics [3], robotics [7,8], exoskeletons [9], energy harvesting [10], morphing skin [11], smart window [12], and smart fabrics [1,13,14]. Typically, the TCPAs are composed of a heating source and a coiled polymer, which is mainly formed by twist insertion of a monofilament fiber and then thermally annealing it to prevent from being untwisted. When uniaxially pre-stretched under load, heating induces the TCPAs to lift the load owing to their large contractile deformation in response to the anisotropic thermal expansion of the polymer fibers in the radial direction.

For studying thermo-mechanical deformation behavior of the TCPAs, researchers have adopted diverse heat sources, such as water, light, ambient air, and electric power [15,16,17]. Among these heating methodologies, particularly, electric power has been widely utilized due to its advantages over others in miniaturization and controllability. For the electric heating that occurs by electric current flowing through a conductor integrating onto the coiled polymer, diverse conductors have been employed in the form of a thin electrode, as well as a metallic wire (e.g., steel, copper) or a silver-plated polymer wire. The electrodes are formed on the surface of the coiled structure by not only painting silver paste [1,18] or conductive elastomer containing silver particles [19], electroless silver plating via chemical reduction of silver ions [20], and spray-coating silver nanowires (AgNWs) [21], but also consecutively winding a carbon nanotube (CNT) sheet drawn from a vertically-grown CNTs on a polymeric fiber [1]. In parallel, the conductive wire is formed by coiling with the polymeric fiber via twist insertion after winding it on the polymeric fiber [1,22,23,24]. These approaches have contributed to eliciting electro-thermally-induced contractile stroke from the TCPAs as providing benefits with respect to temperature controllability, stretchability and processability. However, the electric heating capability of the electrode for the TCPAs has been demonstrated by only implementing a few cyclic actuations [18,19]. In the case of the conductive wire, an achievable tensile stroke was limited to less than 10%, accompanied by slow heating response of the TCPA due to sparse coiling density of the wire [19,24]. Therefore, for practical use, there are still technical challenges in imparting mechanical robustness and electrical stability to the conductors during highly repetitive heating-cooling cycles and simultaneously reducing complexity in fabrication.

Here, we propose a novel and facile approach to chemically form a thin, mechanically-robust and electro-thermally-stable gold (Au) electrode on a twisted and coiled nylon-6 fiber (TCN) with a three-dimensional structure as exploiting gold-sulfur (Au-S) interaction achieved via surface modification chemically-attaching (3-mercaptopropyl)trimethoxysilane (MPTMS) onto the TCN. In this paper, we report studies on not only fabrication, chemical analysis, and mechanical and electro-thermal heating characteristic with morphological observation of the metallic electrodes, but also electro-thermally-induced actuation performance of the TCN actuators (TCNAs) during a large number of repetitive heating-cooling cycles.

## 2. Materials and Methods

### 2.1. Materials

Nylon-6 monofilament fishing line (Tournamenter SE No.3) with a diameter of 285 µm was purchased from Toray Industries, INC (Tokyo, Japan). Nylon-6 film with a thickness of 100 µm were purchased from Goodfellow Cambridge Ltd. (Huntingdon, United Kingdom), respectively. (3-mercaptopropyl)trimethoxysilane (MPTMS, 95%), ethyl alcohol (>99.5%), acetone (>99.5%), and isopropyl alcohol (IPA, 99.5%) were purchased from Sigma-Aldrich (St. Louis, MO, USA). Silver nanowires (AgNWs) solution (AgNWs were dispersed in IPA with a concentration of 0.5 wt %) was purchased from DS Hi-Metal (Ulsan, Korea). The AgNWs have average diameter of 40 nm and length of 20 µm. Deionized water (DI-water) with a resistivity of 18.2 MΩ·cm at 25 °C was achieved from a filtration system of Milli-Q MerckMillipore (Burlington, VT, USA).

### 2.2. Fabrication of a Twisted and Coiled Nylon-6 Structure

Twisted and coiled nylon-6 fibers (TCNs) were fabricated by twist insertion and thermal annealing processes sequentially as shown in Figure 1. By exploiting the fabrication process reported [1], the upper end of the nylon-6 fiber (length: 1200 mm) was clamped to the shaft of a motor and then a load of 19.4 MPa was applied to the bottom end of the fiber by attaching a weight of around 126 g. After inserting twist to the fiber as operating a rotational motor with a consistent speed of 350 rpm, we achieved the TCNs (length/diameter: around 270 mm/670 μm) with a consistent spring index of 2.35 (deviation: < 0.02). After uniaxially stretching to 10% and clamping with a metallic fixture at both ends, the TCN was thermally-annealed in a VDL23 vacuum oven of Binder (Tuttlingen, Germany) at 170 °C for 100 min.

### 2.3. Surface Modification of the TCN

The prepared TCN was precleaned by sequentially immersing it in acetone and IPA bath with sonication for 3 min and then drying at 60 °C in a vacuum oven for 1 h. The precleaned TCN was mechanically stretched to 40% and clamped to a couple of metallic fixtures because the stretching, which is much less than its elastic limit, could sufficiently provide space among the coiled structure stuck to each other for facilitating contact of UV/Ozone plasma onto its overall surface area. After placing the TCN at 10 cm distant from the UV lamp (wavelength: 254 nm) in a chamber of a UVC 300 UV/Ozone system of YUIL (Incheon, Korea), the surface of the TCN was exposed to UV light for 15 min at atmospheric pressure. The UV/Ozone treated TCN immediately immersed into 50 mM solution of MPTMS and deionized water, which has been reported as an effective concentration to moderately hydrolyze MPTMS molecules in water [25], and then it was kept in the solution under magnetic stirring at 40 °C for 30 min. Finally, the MPTMS treated TCN was sonicated in an ethanol bath to remove residual MPTMS and dried using nitrogen gas.

### 2.4. Construction of Metallic Electrode on the TCN

Gold nanoparticles were coated on the surface of both a MPTMS treated and a pristine TCN under the same pre-stretch condition by a Q300TD sputter coater of Quorum Technology (Lewes, United Kingdom) as setting the coating thickness to be 50 nm under a stage rotation with a constant speed. AgNWs were coated on the pristine TCN by a HP-TR1 spray gun of IWATA (Cincinnati, OH, USA). For the AgNWs coating, a frame holding the TCN was fixed on a motorized zig that can control rotational speed. 1 mL of the AgNWs solution was sprayed on the surface of the TCN as simultaneously rotating the zig with constant speed and then dried at room conditions.

### 2.5. Characterization

The surface characteristic of the nylon-6 material before after UV/Ozone plasma treatment was investigated by measuring water contact angle (WCA) via a DSA 25S drop shape analyzer of KRÜSS (Hamburg, Germany). For the measurement, in substitution for the TCN, we used a nylon-6 film with thickness: 350 µm. The morphological characteristic was measured by a Sirion 600 field emission scanning electron microscope (FESEM) of FEI (Hillsboro, OR, USA) and an Axico Sope A1 optical microscope of Carl ZEISS (Stuttgart, Germany). The chemical analysis for the surface modification with MPTMS was performed by a K-Alpha X-ray photoelectron spectroscopy (XPS) of Thermo Fisher Scientific Inc. (Waltham, MA, USA) with 0.1 eV scanning step. Electrical resistance of the metallic electrode was measured by a 34450A digital multimeter of Keysight technologies (Santa Rosa, CA, USA).

### 2.6. Performance Test

Electro-thermal actuation performance of the TCPAs with three different electrodes were evaluated via a performance measurement system composed of a B2901A precision source/measure unit of Keysight technologies, a LK-HD500 laser displacement sensor of Keyence (Osaka, Japan), and a PI 640 thermal imaging camera of Optris (Berlin, Germany), which is shown in Figure 2. In the measurement system, the precision source/measure unit was used to provide an input voltage for electro-thermal heating as measuring an electric current. During the electro-thermal excitation, actuation response and temperature distribution for the TCPAs were simultaneously achieved from the laser displacement sensor and the thermal imaging camera.

## 3. Results and Discussion

### 3.1. Surface Modification of TCN

The TCN was prepared by following the process described in Section 2.2. In order to establish a compliant electrode on the surface of the TCN, we implemented surface modification of the TCN with MPTMS that can lead to strong gold-sulfur (Au-S) covalent bonds between gold nanoparticles and thiol groups in MPTMS molecules. Figure 3 shows a scheme to chemically-attaching MPTMS molecules onto the surface of the TCN. At first, we investigated influence of the plasma treatment on surface characteristic of the nylon-6 material via measuring change in WCA responding to the plasma treatment. Here, we note that the test was performed by using a nylon-6 film instead of a TCN because the TCN with a small fiber diameter, which is much less than 1 mm, has a difficulty in exploiting sessile drop method. Prior to the test, the nylon-6 film was precleaned by the same process implemented before surface modification of the TCN. Figure 4 shows change in the WCA of the nylon-6 film depending on exposure time of UV/Ozone plasma. In the case of a pristine nylon-6 film, the water droplet on the surface formed a contact angle of 94.2°, indicating that the polymer intrinsically possesses hydrophobic nature. However, the surface turned into hydrophilic after exposure to UV/Ozone plasma. The hydrophilicity became enhanced as exposure time of UV/Ozone plasma increased. Particularly, implementation of the plasma treatment as long as 15 min changed the surface close to superhydrophilic, resulting in reduction of the WCA as low as 12.0°. It suggests that the UV/Ozone plasma treatment can be an effective methodology to modify nylon-6 surface with abundant hydroxyl and carboxyl moieties, which has been known as oxygen based polar groups enhancing hydrophilicity [26,27].

Secondly, in order to confirm the chemical surface modification of the nylon-6 film with MPTMS molecules, we investigated XPS spectra for the film before and after MPTMS treatment. In Figure 5a, the full scale XPS spectra revealed that both nylon-6 films have two common peaks at the binding energy of 533 and 285 eV, which are individually assigned to O 1s and C 1s. On the other hand, the MPTMS treated nylon-6 film exhibited distinguishable peaks for Si 2p and S 2p at each binding energy of 102.1 and 162.8 eV. As shown in Figure 5b, the binding energy peak for S 2p can be deconvoluted into two peaks for spin-orbit doublets of S 2p_3/2_ and S 2p_1/2_, corresponding to 162.5 and 163.8 eV, respectively. The binding energy peaks for S 2p_3/2_ and S 2p_1/2_ exhibited a 2:1 area ratio of the peaks with their deviation of 1.3 eV. In parallel, as shown in Figure 5c, a high resolution XPS spectra for Si 2p revealed a binding energy peak at 102.1 eV, which is assigned to Si-O-C bond, indicating that the surface of the nylon-6 film was modified to MPTMS molecules with thiol groups via condensation reaction of silanol groups with hydroxyl groups on the film surface [28,29]. In addition, based on the elemental concentration data taken from XPS survey spectra (Table 1), we also confirmed that MPTMS treated nylon-6 film has remarkably higher Si and S contents than the pristine nylon-6 film.

### 3.2. Mechanical Robustness of Metallic Electrode on the TCN

By implementing chemical surface modification, we achieved the MPTMS chemically-attached TCN, which is termed as MPTMS-TCN. Since thiol groups on MPTMS molecules could form strong covalent bonds with gold nanoparticles, we investigated influence of the gold-sulfur (Au-S) interaction on securing mechanical robustness and stable electrical property of electrode on the TCN. For the study, we established each metallic electrode on the surface of the TCNs by not only sputtering gold nanoparticles on both a pristine TCN and a MPTMS-TCN, but also spray-coating AgNWs solution on a pristine TCN. Using a commercially-available 3M adhesive tape, we performed manual adhesion tests of the electrodes on the TCNs by following a stepwise process of fixing the electrode coated TCN on a glass substrate, attaching the adhesive tape, and peeling the tape off after pressurized rubbing it several times. In addition, as repeating the peel tests, we also measured change in electrical resistance with respect to initial resistance (R/R_0_) for the electrodes. As shown in Figure 6, the SEM observation of each metallic electrode coated on the TCNs revealed that regardless of coating methodology and material, each electrode was uniformly constructed on the whole surface area of the TCN without any noticeable defect site. However, in spite of the uniform coating, the metallic electrodes have crucial difference in mechanical robustness against adhesion force (Figure 7). In the case of the physically coated Au and AgNWs electrodes (Figure 7a,b), the metallic particles were easily peeled off from the surface of the TCN even under soft finger touch. The mechanical loss in the electrode became significant after the adhesion test using the 3M adhesive tape. On the other hand, unlike physically coated electrodes, we rarely observe mechanical loss in the Au electrode on the MPTMS-TCN even after the same adhesion test (Figure 7c). Thanks to the excellent mechanical robustness, the Au electrode on the MPTMS-TCN only maintained stable electrical resistance (R/R_0_ < 2) even after fifteen peel tests at the same area, while the physically coated electrodes easily lost their conductive characteristic in a few peel tests (Figure 7d). We believe that the electrical stability together with the excellent mechanical robustness against the adhesive force can be crucial evidence to support strong Au-S interaction enabled by the chemical surface modification.

### 3.3. Electro-Thermally-Induced Deformation Behavior of the TCNAs

The TCNAs were prepared by constructing the metallic electrode on the surface of the TCN. According to materials for electrode and surface modification for the TCN, the TCNAs were classified as TCNA I, TCNA II, and TCNA III. The TCNA I was prepared by coating Au nanoparticles on the surface of the MPTMS-TCN. Unlike the TCNA I, the TCNA II and III were based on the pristine TCN. The sputtered Au nanoparticles and the spray-coated AgNWs were used as the electrode for TCNA II and III, respectively. Both the Au and AgNWs electrodes exhibited a fairly consistent initial electrical resistance (R_i_) of 32 ± 1.6 Ω and 95 ± 3.8 Ω, respectively. There was no meaningful difference in the R_i_ of the Au electrodes whether the surface of the TCN was modified or not. Even after pressurized clamping of each TCNA with a couple of rigid frames and applying a load of 100 g, only small increase in their electrical resistance (R/R_i_: < 5%) was observed, indicating that the metallic electrodes could secure stable electrical property before the actuation test. Under the same loading condition, we investigated electro-thermally-induced deformation behavior of the TCNAs by simultaneously measuring their contractile strain and surface temperature using the performance evaluation system, which is shown in Figure 2.

Figure 8a shows heating temperature dependent contractile strain of the TCNAs. Here, the contractile strain is defined as decrease in length of the TCNA divided by its length after loading, which is expressed as ∆L/L_loading_ × 100 (%). When the heating temperature increased as high as 150 °C, the contractile strain of all TCNAs increased quadratically, while their temperature-strain curves were clearly different with the electrode. Based on curve fitting of the experimental data, we confirmed that actuation behavior of the TCNAs could follow simple exponential functions (R-square ≈ 0.98) as representing a strong dependency on heating temperature. Meanwhile, the TCNAs with Au electrode (TCNA I and II) exhibits an analogous temperature-strain profile, producing the contractile strain as large as 16.8% with a sensitivity of 0.14%/°C and its high reproducibility (Pearson correlation coefficient: 0.983), regardless of the surface modification for chemically-attaching the MPTMS molecules onto the TCN, which can lead to enhancement of adhesion force between Au nanoparticles and the TCN. Unlike TCNA I and II, the TCNA with AgNWs electrode (TCNA III) could only produce a strain as high as 12%, which is 27% lower than that of the TCNA I and II, with a sensitivity of 0.1%/°C and relatively low reproducibility (Pearson correlation coefficient: 0.979), although the TCNAs consistently heated to 150 °C. It suggests that the use of AgNWs electrode cannot fully elicit an achievable stroke from the TCNA.

As shown in Figure 8b, the quadratic increasing tendency of the strain was consistently observed during repetitive actuation tests using five TCNAs each of the three forms (TCNA I, II, and III) at four different temperature conditions. The reachable magnitude of strain for each type of the TCNAs was also almost identical, revealing only a small deviation of strain, which is less than 0.4%. It indicates that our fabrication process can allow highly repetitive production of the TCNAs with a consistent performance. In parallel, to find the reason for electrode dependency in actuation performance, we monitored temperature distribution on the surface of the TCNAs via a thermal imaging camera during electro-thermal heating to 150 °C. As shown in Figure 8c, the Au electrode allows the TCNA to be heated with a fairly consistent temperature in whole surface area with a small deviation in temperature at the local surface areas, which is less than 7%. On the other hand, the AgNWs electrode exhibited area-dependent temperature distribution with a large deviation in temperature as high as 20%. The uneven temperature distribution can be strongly correlated with irregularity in coating density of AgNWs, which causes electrical resistance to be much different with area. As the result, presence of the underheated areas led to the reduction in stroke at the localized areas and it resulted in degradation of overall strain of the TCNA III. In addition, the temperature distribution affects recovery from the deformed state during a heating-cooling cycle. As compared to the TCNA I, the TCNA III exhibited relatively larger hysteresis loop during a heating-cooling cycle (Figure 8d), indicating that building a compliant electrode enabling uniform and sustainable electro-thermal heating is important to impart highly reversible actuation to the TCNA as suppressing hysteresis behavior.

### 3.4. Durability Test of the TCNAs

Based on comparative study of electro-thermally-induced actuation performance of the TCNAs, we found that as compared to the AgNWs electrode, the Au electrode could contribute to eliciting larger tensile stroke with relatively small hysteresis during a heating-cooling cycle regardless of level in adhesive force between Au nanoparticle and TCN.

In order to investigate influence of the strong Au-S interaction on actuation durability, we monitored change in contractile strain of the TCNAs during repetitive actuations over 1000 heating-cooling cycles. For the test, we operated the TCNAs under the loading of 100 g by repeating an actuation cycle composed of electro-thermally heating to 150 °C, keeping for 20 s and then naturally cooling down for the same period of time. Figure 9 shows the result of durability test. During first 500 cycles, we consistently observed a decreasing tendency of contractile strain for all TCNAs. In the case of TCNA II and III, the contractile strain was rapidly reduced to around 40% as compared to the value at initial cycles. When the number of actuation cycles reached to over 1000, both TCNAs completely lost their actuation capability. On the other hand, thanks to surface modification of TCN enabling strong Au-S interaction, TCNA I exhibited stable contractile stroke without any significant performance degradation after 15% reduction in amplitude of contractile strain during first 500 actuation cycles. Moreover, the TCNA shows excellent actuation durability as maintaining a large and repeatable tensile stroke with amplitude of contractile strain over 12% even after 2000 cyclic actuations.

Meanwhile, for clearly understanding the reason of difference in actuation performance of the TCNAs, we monitored change in temperature distribution on the TCNAs as increasing the number of actuation cycles to over 1000. As shown in Figure 10, when the number of actuation cycles reached 1000, the electrodes exhibited clear difference in electro-thermal heating performance with respect to heating capability and uniformity in temperature distribution. In the case of TCNA II and III, we observed that the local areas heating much below the set temperature gradually increased during repetitive cyclic actuations and their uneven temperature distribution became significant, accompanied by ∆T_h_ over 80 °C, as the number of cyclic actuations was closed to 1000. It caused degradation of overall tensile stroke. Here, ∆T_h_ is defined as the difference between the maximum and the minimum temperature in whole length of the TCNA. Particularly, at the specific area overheated to the temperature over 170 °C, the coiled structure began to irreversibly loosen and then fully unraveled, resulting in complete loss of actuation capability. Based on the SEM observation, we also found that the electrodes at the overheated area was significantly damaged, suffering from micro-cracks as well as partially peeled off from the TCN structure. It indicates that both physically coated electrodes hardly possess strong adhesion to TCN for resisting a large and repetitive electro-thermal deformation.

Unlike TCNA II and III, the TCNA I maintained not only consistent temperature distribution on its whole surface area with a remarkably small ∆T_h_ (<10 °C), but also smooth electrode surface similar to its initial state without any significant physical damage although 16% decrease in attainable temperature was unavoidable during first 500 heating-cooling cycles. The result suggests that the strong Au-S covalent bonds between gold nanoparticles and MPTMS-TCN can remarkably contribute to constructing a highly compliant electrode that secures high robustness against repetitive electro-thermally-induced deformation. Video S1 is a demonstration of the actuation performance of multiple TCNA I that is capable of producing contractile strain of 16.8% as lifting a load of around 1000 g in response to electro-thermal heating.

## 4. Conclusions

In summary, we developed a novel and facile methodology to chemically construct a thin and highly compliant electrode for a TCNA with three-dimensional structure via Au-S interaction in a formation of covalent bonds between gold nanoparticles and thiol groups on MPTMS-TCN, which was achieved from surface modification chemically-attaching the MPTMS molecules on the surface of the TCN. Based on analysis of XPS spectra, we presented clear evidence of the surface modification of the nylon-6 with MPTMS molecules via condensation reaction of hydrolyzed MPTMS molecules with hydroxyl groups on UV/Ozone plasma treated nylon-6. By exploiting repetitive adhesion tests, we demonstrated that the Au electrode established on the MPTMS-TCN could secure excellent mechanical robustness as simultaneously maintaining stable electrical resistance (R/R_0_ < 2) even after fifteen peel tests. Due to the benefit from the strong Au-S interaction, the compliant electrode not only enabled highly repetitive and uniform electro-thermal heating on the whole surface of MPTMS-TCN with only a small difference in temperature, which is less than 7%, but also could elicit a large and repeatable contractile strain over 12%, lifting a load of 100 g from the TCNA even during 2000 heating-cooling cycles.

Our future study will include finding ways to elicit contractile strain of the TCNA analogous to mammalian skeletal muscle with reversible tensile stroke over 20%. We also need to investigate not only TCNA with cost-effective metals instead of gold as the compliant electrode, but also integration of stretchable strain sensor onto the TCNA enabling real-time control in tensile stroke for practical use in promising applications such as soft robots and wearable human-assistive devices.

## Figures and Tables

**Figure 1 polymers-14-03601-f001:**
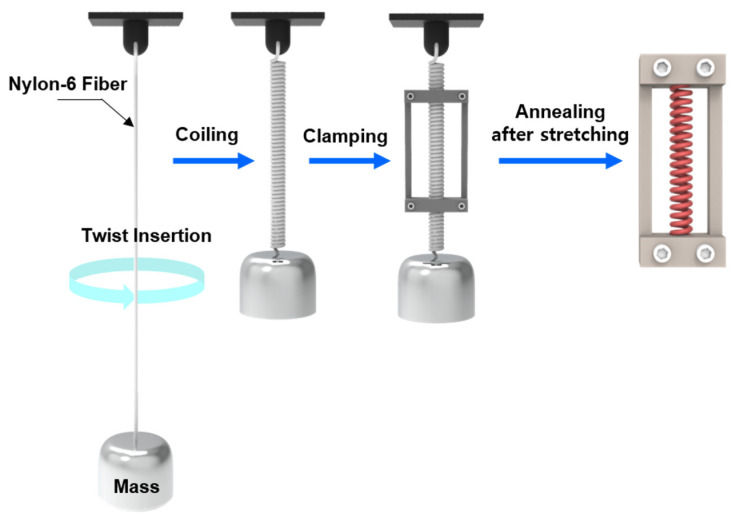
An illustrated fabrication process of the TCN structure.

**Figure 2 polymers-14-03601-f002:**
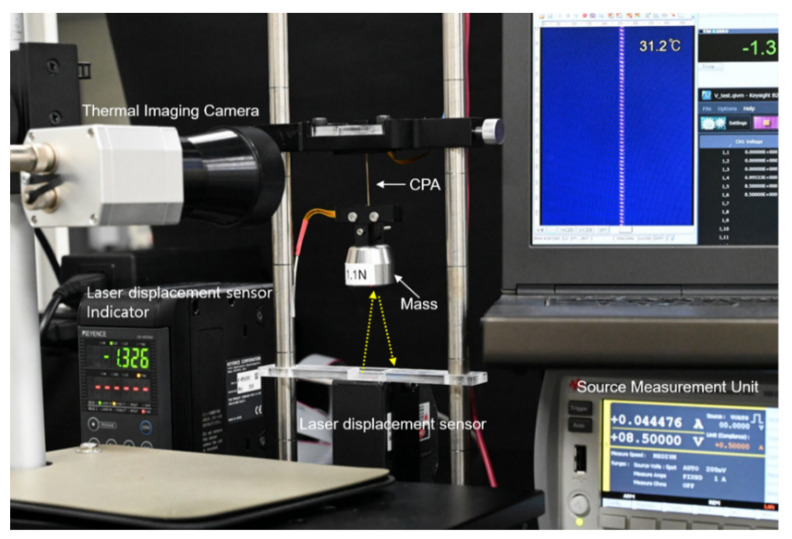
A photograph of performance evaluation system for the TCNA.

**Figure 3 polymers-14-03601-f003:**
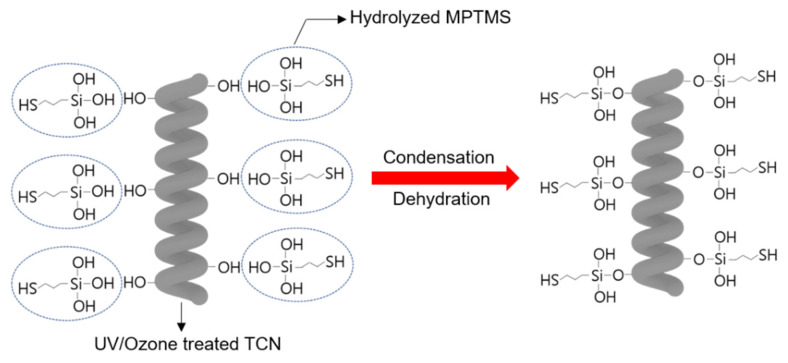
A schematic illustration of chemically-attaching MPTMS molecules onto the surface of a TCN.

**Figure 4 polymers-14-03601-f004:**
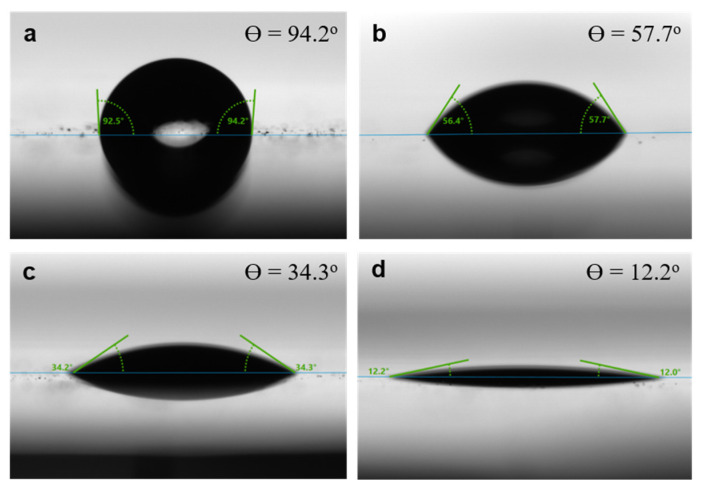
Change in the WCA of the nylon-6 films according to exposure time of UV/Ozone plasma: (**a**) without exposure, (**b**) 3 min, (**c**) 5 min, and (**d**) 15 min.

**Figure 5 polymers-14-03601-f005:**
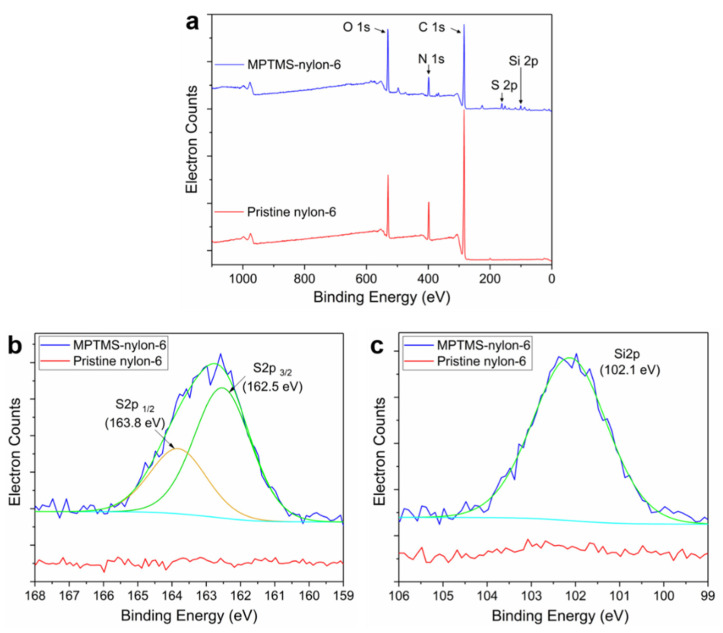
Comparison of XPS spectra of a pristine nylon-6 and a MPTMS-nylon-6 film: (**a**) full-scale spectra, (**b**,**c**) high-resolution spectra for S 2p and Si 2p. In the overlapped lines, the light blue and green line are repectively assigned to XPS spectra experimentally measured and after smoothing.

**Figure 6 polymers-14-03601-f006:**
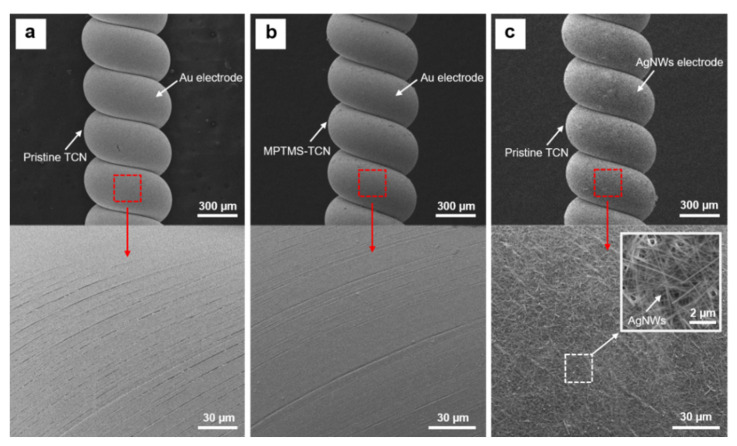
SEM surface images of metallic electrodes formed on the TCN: (**a**) Au electrode on a pristine TCN, (**b**) Au electrode on MPTMS-TCN, and (**c**) AgNWs electrode on pristine TCN.

**Figure 7 polymers-14-03601-f007:**
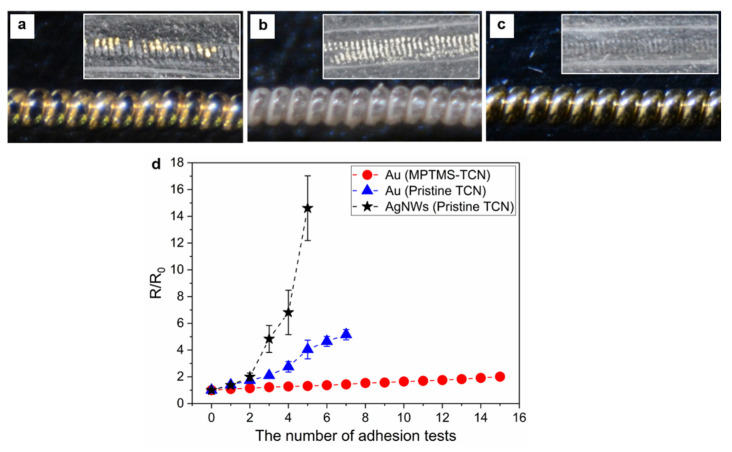
(**a**-**c**) Optical microscope images of metallic electrode on each TCN after adhesion test: (**a**,**b**) Au and AgNWs electrode on a pristine TCN and (**c**) Au electrode on MPTMS-TCN. (**d**) Comparison of change in electrical resistance with respect to initial resistance during repetitive peel tests. Each inset is photograph showing surface of adhesive tape peeled off from each electrode.

**Figure 8 polymers-14-03601-f008:**
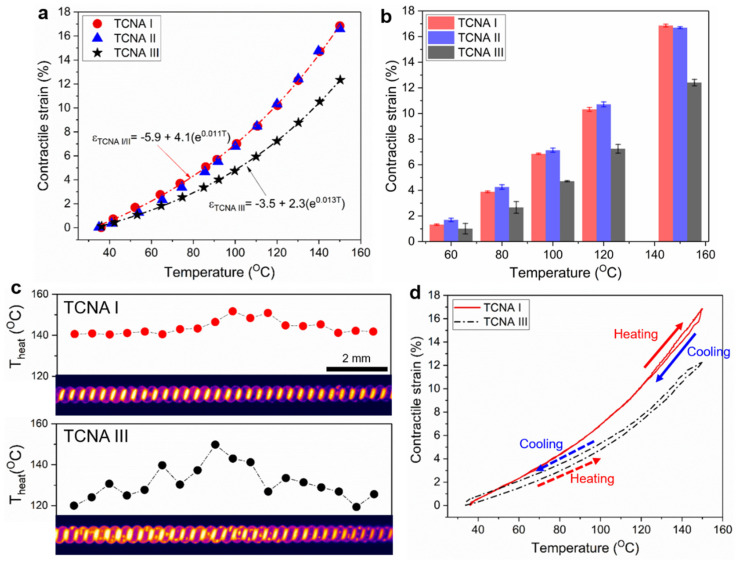
Electro-thermally induced actuation performance of the TCNAs with different electrode: (**a**) comparison of temperature-strain profiles of the TCNAs; (**b**) comparison of contractile strain with deviation measured at four different temperatures for five TCNAs each of TCNA I, II, and III; (**c**) thermal images with temperature distribution of the TCNAs heated to 150 °C; and (**d**) their hysteresis curves of the actuation performance during a heating-cooling cycle.

**Figure 9 polymers-14-03601-f009:**
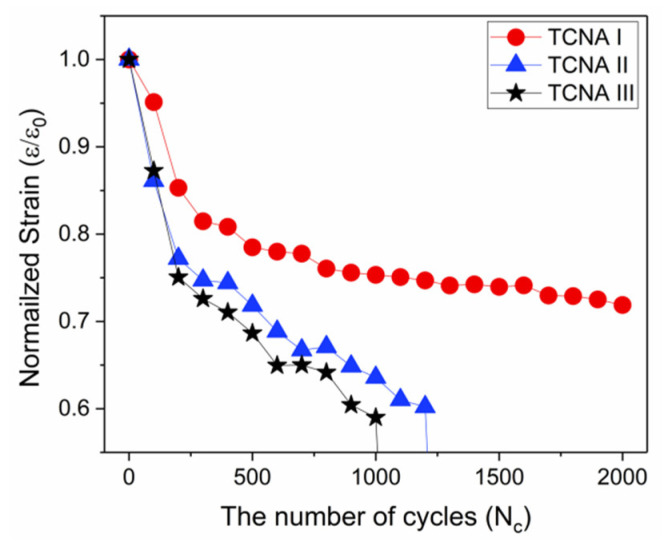
Comparison of electro-thermally induced contractile strain of the TCNAs during repetitive cyclic actuations.

**Figure 10 polymers-14-03601-f010:**
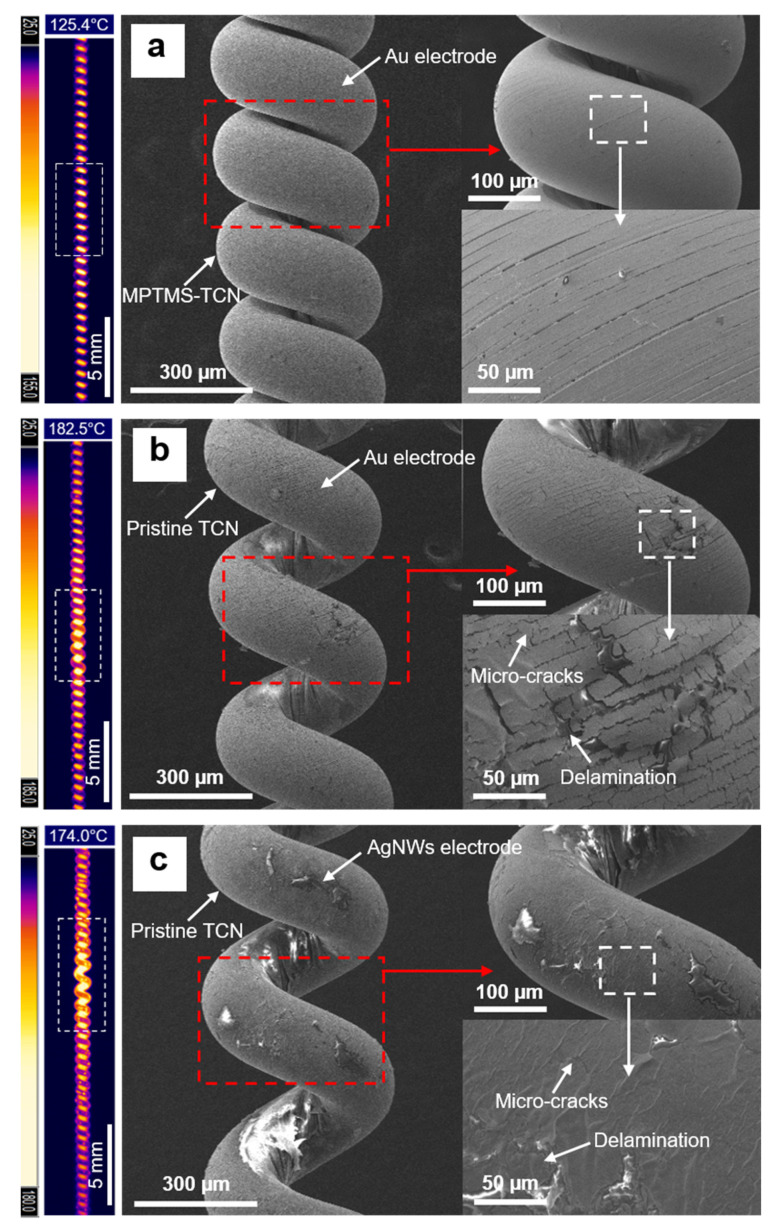
Thermal images with their maximum temperature and SEM surface images of the TCNAs after 1000 cyclic actuations: (**a**) TCNA I, (**b**) TCNA II, and (**c**) TCNA III.

**Table 1 polymers-14-03601-t001:** Elemental composition of a pristine nylon-6 and a MPTMS-g-nylon-6 film achieved via XPS.

	C 1s	O 1s	N 1s	Si 2p	S 2p
Pristine nylon-6	75.87	11.63	12.05	0.26	0.18
MPTMS-nylon-6	65.70	20.17	8.54	3.22	2.38

## Data Availability

The data presented in this study are avaiable on request from the corresponding author.

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
