# Peer review of "A Thermo-Mechanically Robust Compliant Electrode Based on Surface Modification of Twisted and Coiled Nylon-6 Fiber for Artificial Muscle with Highly Durable Contractile Stroke"

_polymers, 2022, doi:10.3390/polym14173601_

Round 1

Author Response

Review comments

Reviewer #1 (Comments to the Author):

This paper proposed the surface treatment method before coating gold particles on the surface of twisted and coiled nylon-6 fiber actuator (TCNA). The proposed actuator kept working over 2000 cyclic actuations, though the conventional actuators lost working about 1000 cyclic actuations. Please check the following points:

[Major points]

1. In Fig.8. Please show the reproducibility of experimental results. Because it is difficult to make the same twisted and coiled actuators, they may show the scattering in the experimental results. It should discuss the reproducibility of experimental results.

[Authors]: Thank you for kind comments. As carefully considering reviewer’s comment, we prepared fifteen TCNs by following the fabrication process described in section 2.2. Using the TCNs, we constructed five TCNAs each of the three forms denoted as TCNA I, II, and III. First, we note that our fabrication process allows the TCNs to possess a consistent spring index of 2.35 with their small difference (deviation: <0.02), indicating that the coiled structure of the TCNs can be almost identical. Moreover, as described in section 3.3, we already investigated electrical resistance of the five TCNAs each of the three forms. Based on the study, we confirmed that the Au and AgNWs electrode formed onto the TCNs could secure a consistent electrical property with small deviation, which is less than 5 %.    

For performance evaluation of the five TCNAs each of the three forms, we measured their contractile strain at four different temperature conditions corresponding to 60°C, 80°C, 100°C, and 150°C. As the result shown in Figure 8b of our revised manuscript, each type of the TCNAs exhibited some deviation of the reachable strain depending on the samples. As compared to the TCNA I and II with sputtered Au electrode, the TCNA III with AgNWs electrode exhibited relatively higher deviation of the contractile strain. However, we note that their deviation of the contractile strain for the TCNA I and II was less than 0.2% regardless of heating temperature. Even in the case of TCNA III, their deviation was still below 0.4%. Based on the tests, we confirmed that our fabrication process could allow repeatable production of the TCNAs with consistent actuation performance. In our revised manuscript, we added not only spring index with deviation for the TCN in section 2.2, but also Figure 8b with figure caption for the test result and description for the result in section 3.3. The modified parts were highlighted by using yellow color.

2. There seems to be no evidence that the strong covalent bonds of gold-surfur (Au-S) were formed on the electrode.

[Authors]: Thank you for kind comments. Over the last decade, strong interaction of sulfur with gold as a formation of gold-sulfur (Au-S) covalent bonds has been proved and it became a well-known phenomenon [A,B]. Recently, peel test with adhesive tape has been introduced as a methodology to demonstrate presence of the chemically-induced strong interaction on metal-sulfur interface [C]. As adopting the methodology reported, we carried out microscopic surface observation after peel test for visualized demonstration of the chemically-induced strong adhesion between Au electrode and MPTMS-TCN.

As carefully considering reviewer’s comment, in order to suggest a supplementary evidence, we additionally investigated changes in electrical resistance with repect to initial resistance (R/R0) of three different electrodes formed onto the TCN in response to the number of peel tests with adhesive tape. For the test, we used the five TCNAs each of the three form. As the result, the Au electrode on the MPTMS-TCN maintained stable electrical resistance (R/R0 <2) even after 15-repetition adhesion tests at the same area, while the physically coated electrodes easily lost their conductive characteristic in a few peel tests. We believe that the electrical stability together with the excellent mechanical robustness against the adhesive force can be a crucial evidence to support strong Au-S interaction enabled by the chemical surface modification. In our revised manuscript, we added a graph showing the result as Figure 7d with the figure caption and description for the result in section 3.2. The modified part was highlighted by using yellow color.

[A] Y. Xue, X. Li, H. Li, and W. Zhang, Quantifying thiol-gold interactions towards the efficient strength control, Nat. Commun. 5, 4348, 2014.

[B] E. Pensa, E. Cortes, G. Corthey, P. Carro, C. Vericat, m.h. Fonticelli, G. Benitez, A.A. Rubert, and R.C. Salvarezza, Acc. Chem. Res. 45, 1183, 2012.

[C] Y.-T. Kwon, Y.-S. Kim, Y. Lee, S. Kwon, M. Lim, Y. Song, Y.-H. Choa, and W.-H. Yeo, ACS Appl. Mater. Interfaces 10, 44071, 2018.

3. The same explanation about experimental methods can be seen in Section 2 and 3. Please edit the manuscript clearly and concisely.

[Authors]: Thank you for kind comment to improve conciseness of our manuscript. As considering reviewer’s comment, we carefully reviewed the manuscript and then we modified the manuscrpt by both eliminating the duplicated explanation from the section 3 and transfering parts of the explanation in section 3 to section 2. The modified part was highlighted by using yellow color.

[Minor points]

1. In Fig.6, 8(b) and 10. Please insert the scale to identify the length.

[Authors]: Thank you for kind comments. As considering reviewer’s comment, we inserted the scale on the bar at each SEM image in Figure 6 and Figure 10. For Figure 8b (Figure 8c in revised manuscript), we also inserted the scale to indentify the length of TCNA in the thermal image.

2. In Fig.8(a). Please show the definition of contractile strain.

[Authors]: Thank you for pointing our missing information. As considering reviewer’s comment, we added a sentence for the definition of contractile strain to section 3.3. The modified part was highlighted by using yellow color.

3. In Fig.8(c). Please show which heating and cooling path are

[Authors]: Thank you for kind comments. As considering reviewer’s comment, we expressed the heating and cooling path using a couple of arrows with text in Figure 8c (Figure 8d in revised manuscript).

Reviewer 2 Report

The manuscript is very well written and presented. I recommend for publication after the incorporation of minor changes.

1. Introduction is very short and The state-of-the-art comparisons for the proposed work are missing in this paper. Add more (10-15) articles and then do a critical analysis of previous research. State explicitly the shortcomings of previous research. 

2. In Materials & Method, provide a source of raw materials and their purity level. How did you choose the experiment setting? Elaborate experiments parameters

3. Is there any standard adopted to fabricate the samples? What is the standard used to perform the Electro-thermal actuation performance test?

4. Highlight the observations and elements in the SEM images.

5. Highlight the novelty of your methodology.

6. The biggest shortcoming of the research is that there is no analysis of errors, analysis of sensitivity of results and analysis of uncertainty of results.
7. The Conclusion section should be rewritten. Highlight your scientific contribution. Highlight the benefits of your research. Define shortcomings and future research.

Author Response

Review comments

Reviewer #2 (Comments to the Author):

The manuscript is very well written and presented. I recommend for publication after the incorporation of minor changes.

1. Introduction is very short and the state-of-the-art comparisons for the proposed work are missing in this paper. Add more (10-15) articles and then do a critical analysis of previous research. State explicitly the shortcomings of previous research;

[Authors]: Thank you for kind comments. As carefully considering reviewer’s comment, we additionally cited twelve references together with critical analysis and also clearly described the shortcomings of previous researches. The modified parts were highlighted by using gray color.

2. In Materials & Method, provide a source of raw materials and their purity level. How did you choose the experiment setting? Elaborate experiments parameters.

[Authors]: Thank you for pointing out missing information. As carefully considering reviewer’s comment, we additionally described the mising information in section 2.1 with respect to source of raw materials and their purity level. Moreover, together with explanation for the experiment setting, we cited two references and also described details in experimental parameters to section 2. In our revised manuscript, the modified part was highlighted by using gray color.

3. Is there any standard adopted to fabricate the samples? What is the standard used to perform the Electro-thermal actuation performance test?

[Authors]: Since Baughman group first reported actuation capability of the twisted and coiled polymers (TCPs) as a new class of artificial muscle [*, ref. [1] in our manuscript], their fabrication process to achieve the TCP structure has currently been considered as a standard. In our study, we construct the twisted and coiled nylon-6 (TCN) structure by exploiting the process reported. Similarly to our performance test methodology, most of researchers evaluate tensile stroke of the TCPAs under loading by measuring change in its length using a laser displacement sensor at the bottom surface of the weight as modulating electrical voltage applied via a metallic wire integrated or electrode formed onto the TCP [*, **]. In parallel, thermal image camera as well as thermocouple has been widely used to measure electrically-modulated temperature.

[*] C.S. Haines et al. Artificial Muscles from Fishing Line and Sewing Thread. Science 2014, 343, 868–872.

[**] M. Suzuki and N. Kamamichi. Displacement control of an antagonistic-type twisted and coiled polymer actuator. Smart Mater. Struct. 2018, 27, 035003.

 4. Highlight the observations and elements in the SEM images.

[Authors]: Thank you for kind comments. In order to highlight the observation and elements in the SEM images (Figure 6 and 10), we modified the figures in our revised manuscript by adding figure legends.

5. Highlight the novelty of your methodology.

[Authors]: Thank you for kind comments. As carefully considering reviewer’s comment, we modified setences described in abstract and the last paragraph of introduction in order to clearly express the novelty in our methodology. The modified parts were highlighted by using gray color.

6. The biggest shortcoming of the research is that there is no analysis of errors, analysis of sensitivity of results and analysis of uncertainty of results.

[Authors]: Thank you for kind comments. As carefully considering reviwer’s comment, we additionally prepared fifteen TCNs by following the fabrication process described in section 2.2. Using the TCNs, we constructed five TCNAs each of the three forms denoted as TCNA I, II, and III. First, we investigated reproducibility of the TCN structure by measuring spring index of fifteen TCNs. Based on the study, we confirmed that our fabrication process allows the TCNs to possess a consistent spring index of 2.35 with their small difference (deviation: <0.02), indicating that the coiled structure of the TCNs can be almost identical. Secondly, using the five TCNAs each of the three forms, we investigated sample dependency on electrical property of three different electrodes formed onto the TCN by measuring changes in their electrical resistance with repect to initial resistance (R/R0) in response to the number of peel tests with adhesive tape. Together with a new Figure (Figure 7d in revised manuscript), we described the results with discussion in section 3.2. Thirdly, we also additionally investigated deviation of the contractile strain by measuring their contractile strain at four different temperature conditions corresponding to 60°C, 80°C, 100°C, and 150°C. Together with a new Figure (Figure 8b in revised manuscript), we described the result including discussion with respect to performance deviation in section 3.3. Finally, in section 3.3, we described regarding sensitivity with Pearson correlation coefficient of TCNA I/II and TCNA III, which was calculated by implementing fitting their temperature-strain curve and the modified part was highlighted by using gray color. 

7. The Conclusion section should be rewritten. Highlight your scientific contribution. Highlight the benefits of your research. Define shortcomings and future research.

[Authors]: Thank you for kind comments. As carefully considering reviewer’s comment, we rewrote the conclusion. The modified part was highlighted by using gray color. 

Round 2

Reviewer 1 Report

The manuscript was revised clearly and concisely.